# Factors influencing implementation and sustainability of interventions to improve oral health and related health behaviours in adults experiencing severe and multiple disadvantage: a mixed-methods systematic review

Deepti A John ![ORCID],[1] Emma A Adams ![ORCID],[1] Laura J McGowan,[1] Emma C Joyes,[1] Catherine Richmond,[1] Fiona R Beyer,[2] David Landes,[3] Richard G Watt,[4] Falko F Sniehotta,[5,6] Martha Paisi,[7] Clare Bambra,[1] Dawn Craig,[1] Eileen Kaner,[1] Sheena E Ramsay[1]

For numbered affiliations see end of article.

**Correspondence to**
Deepti A John;
deepti.john@newcastle.ac.uk

## ABSTRACT

**Objectives** Among people experiencing severe and multiple disadvantage (SMD), poor oral health is common and linked to smoking, substance use and high sugar intake. Studies have explored interventions addressing oral health and related behaviours; however, factors related to the implementation of these interventions remain unclear. This mixed-methods systematic review aimed to synthesise evidence on the implementation and sustainability of interventions to improve oral health and related health behaviours among adults experiencing SMD.

**Methods** Bibliographic databases (MEDLINE, EMBASE, PsycINFO, CINAHL, EBSCO, Scopus) and grey literature were searched from inception to February 2023. Studies meeting the inclusion criteria were screened and extracted independently by two researchers. Quality appraisal was undertaken, and results were synthesised using narrative and thematic analyses.

**Results** Seventeen papers were included (published between 1995 and 2022). Studies were mostly of moderate quality and included views from SMD groups and service providers. From the qualitative synthesis, most findings were related to aspects such as trust, resources and motivation levels of SMD groups and service providers. None of the studies reported on diet and none included repeated offending (one of the aspects of SMD). From the quantitative synthesis, no difference was observed in programme attendance between the interventions and usual care, although there was some indication of sustained improvements in participation in the intervention group.

**Conclusion** This review provides some evidence that trust, adequate resources and motivation levels are potentially important in implementing interventions to improve oral health and substance use among SMD groups. Further research is needed from high quality studies and focusing on diet in this population.

**PROSPERO registration number** CRD42020202416.

## STRENGTHS AND LIMITATIONS OF THIS STUDY

⇒ Comprehensive search strategy was used to gather evidence in this mixed-methods systematic review.
⇒ Consolidated Framework for Implementation Research (CFIR) was used for the data extraction.
⇒ Confidence in the papers was limited due to moderate quality of the papers.
⇒ The included studies were not excluded based on their quality, as they contributed relevant information to this systematic review.

## INTRODUCTION

Severe and multiple disadvantaged (SMD) populations are individuals who have experienced homelessness, substance use, offending or a combination of all three.[1] They experience disproportionately high levels of poor physical and mental health along with high levels of occupational deprivation,[1] which results in isolation and difficulty in accessing healthcare services.[2] There is also an added burden of stigma that affects their access and engagement.[3]

Among people experiencing SMD, oral health problems have been highlighted as one of the major unmet needs.[4] This is aggravated by high levels of smoking, substance and alcohol use and poor diet (high intake of sugar).[4 5] Elevated tobacco use make them more susceptible to periodontal disease, tooth loss, oral lesions and oral cancer.[5 6] Research also shows that they do not meet the daily nutritional requirements and have high levels of sugar consumption.[6 7] Oral health has an overall impact on physical and mental

well-being.[8] It is, therefore, important to address not only oral health concerns in people experiencing SMD but also related health behaviours such as smoking, alcohol and substance use, and poor nutrition.[1 9]

Previous papers focus on intervention design and outcomes, and none focus on the implementation approach of these interventions especially in people experiencing SMD.[10–12] Hence, there is a need for evidence on interventions addressing these health challenges, with a specific focus on ways to improve implementation and long-term sustainability of interventions. Frameworks are used to apply a theoretical underpinning to our understanding of why implementation of interventions succeed or fail. The Consolidated Framework for Implementation Research (CFIR), which is composed of five domains, was used as a theoretical framework to identify the facilitators and barriers that influence implementation.[13 14] This framework, therefore, assists with bridging the gap between research and practice, as well as reducing the challenges of implementing these interventions.[15]

To investigate how we can improve implementation and sustainability, we conducted this systematic review to synthesise various factors such as acceptability, settings and potential adverse effects of interventions that improve oral health and related health behaviours of adults with SMD.

## METHODS
The research protocol was pre-registered and published registered with the Prospective Register of Systematic Reviews (PROSPERO) (reg. no: CRD42020202416).[16 17] The review was reported according to Preferred Reporting Items for Systematic Reviews and Meta-Analyses (PRISMA) guidelines.[18]

### Search strategy
The search strategy (see online supplemental file) was formulated and conducted with an information specialist within the research team. The following electronic databases—MEDLINE (Ovid), EMBASE (Ovid), CINAHL (Ebsco), APA PsycINFO (Ovid) and Scopus—were searched for relevant qualitative, quantitative and mixed-method studies from inception to February 2023. Grey literature searches were conducted using Google Incognito and selected charity organisation websites such as Fulfilling Lives, Crisis and Groundswell, which were informed by the expertise of the research team. Forward and backward citation search of the included studies were also conducted.

### Study selection
The search results were downloaded and deduplicated using EndNote V.20.4.1 and the uploaded into Covidence, an online tool for managing the whole systematic review process.[19] Title, abstracts and full texts were independently screened by two reviewers. In case of a discrepancy, consensus was reached after consultation with a

**Table 1** Eligibility criteria used to select the studies

| Eligibility criteria | |
|---|---|
| Population | Adults aged 18 or above, who experience SMD comprising of either homelessness (rough sleeping or other types of insecure accommodation), repeated offending or frequent substance use that co-occurs with homelessness or repeated offending.[17] Perspectives of staff who work with SMD groups and stakeholders such as policy makers and commissioners. |
| Intervention | Structural, community and individual level interventions.[17] |
| Outcomes | Views from SMD groups and other stakeholders (policy makers, service providers, voluntary sector, etc) about implementation and sustainability of interventions which include acceptability, content, settings, potential harms, uptake and retention.[17] |
| Study design | Qualitative, quantitative and mixed-method studies |

SMD, severe and multiple disadvantage.

third reviewer. Table 1 presents the inclusion criteria used during screening.

### Data extraction and quality appraisal
The data extraction and quality assessment for all the included studies were conducted by one reviewer and cross-checked by a second reviewer. Included studies were critically appraised to guide how much confidences could be placed on the findings. Qualitative studies were appraised using the Critical Appraisal Skills Programme (CASP) qualitative checklist.[20] Quantitative studies were appraised using Cochrane's Risk of bias for randomised controlled trials (RCTs).[21] For cross-sectional studies, the National Institutes of Health (NIH) Study Quality Appraisal Tool was used.[22] Qualitative studies were rated as good, moderate or low quality, which was informed by a scoring system: scores 9–10 was of high quality, 7.5–9 was of moderate quality and <7.5 was of low quality.[23] The scoring was informed by the quality checklists. Studies were not excluded based on their quality; poor reporting is not always reflective of poor methodology.[24] Studies were included on whether they contributed data relevant or novel data to this review.[24] Moreover, including all studies allowed gathering the global evidence related to the review questions.

### Data synthesis
Abstracts and data from the results of included studies were uploaded on to NVivo software (QSR International, Melbourne, Australia, V.12, Release 1.6.1). Narrative synthesis was undertaken. Deductive codes based on the CFIR framework were used to initially code the findings followed by a three-step inductive synthesis process

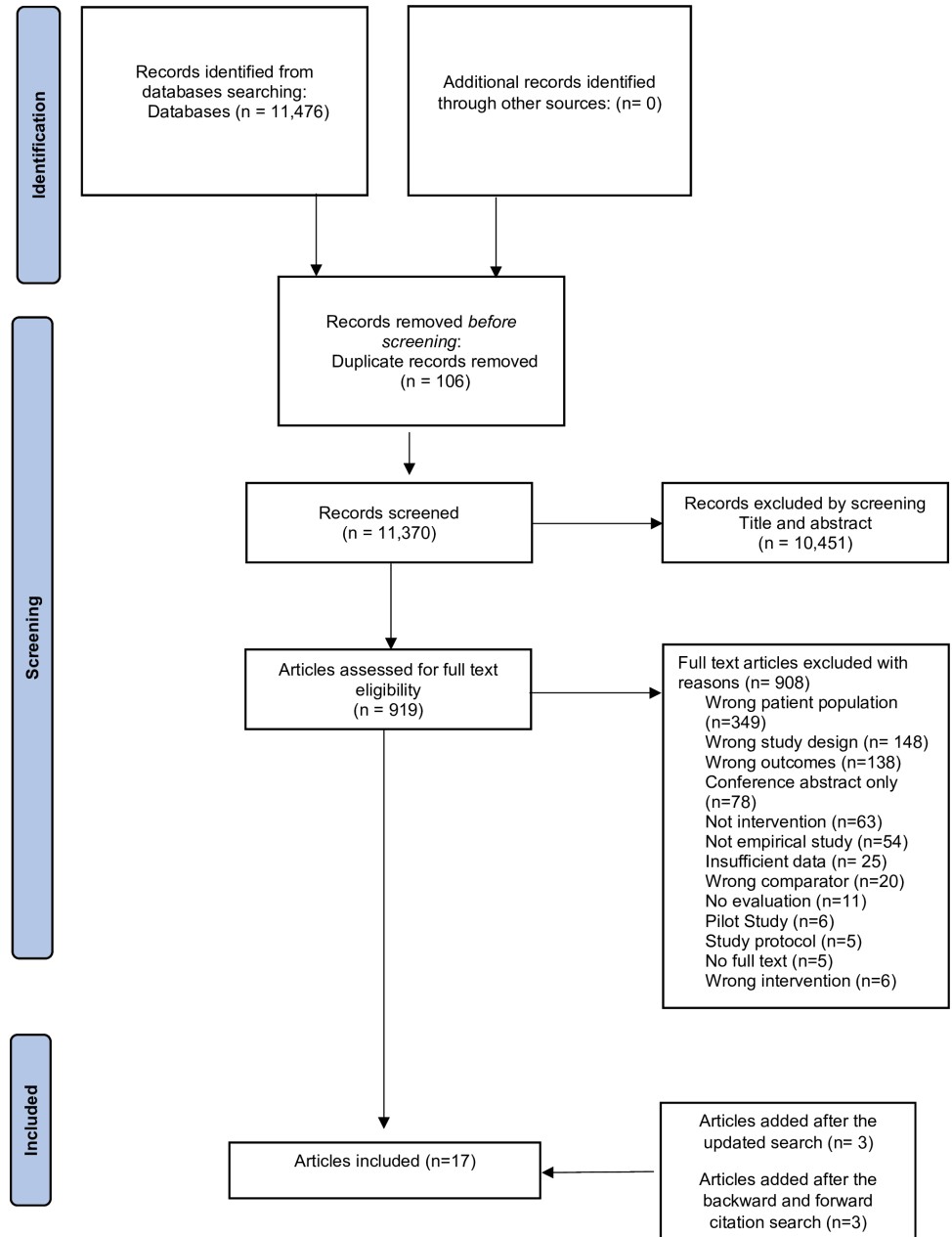

**Figure 1** PRISMA (Preferred Reporting Items for Systematic Reviews and Meta-Analyses) flowchart for the search results.

which involved coding the text, identifying the themes and creating the subthemes. To maximise thematic yield, data reported in different papers but from the same study were individually coded. The developing themes and subthemes were discussed with the other reviewers and consensus was reached regarding these.

## RESULTS

Seventeen articles (12 individual studies) met the inclusion criteria and were included in this systematic review. Figure 1 presents the PRISMA flowchart for the included studies. Table 2 presents the descriptive summaries of the included studies. The papers were published between 1995 and 2022, and were related to interventions targeting oral health,[25–32] substance use[33–39] and smoking, and none on diet.

SMD groups in the studies found in this review included young adults, single mothers, veterans and adults with co-occurring conditions of severe mental illness. Based on the information reported in the studies, most of the interventions were focused on adults who were experiencing homelessness and substance use issues,[33–40] but did not explicitly report on whether they included those who had repeated involvement with the criminal justice system.

### Quality appraisal

Of the 12 articles reporting qualitative findings, 2 were of low quality due to lack of detailed findings and methodology not being reported adequately,[31 36] and 5 were of

**Table 2** Descriptive summaries of the 17 included studies, including quality appraisal (high quality/moderate quality/low quality)

| No. | First author (year) and country | Sample size and age of the participants | Participant group | Intervention description | Type of research, data collection and analysis | Findings of the paper relevant to the review | Quality appraisal/ ROB |
|---|---|---|---|---|---|---|---|
| 1 | Beaton and Freeman[25] (2016) UK | n=20, age not mentioned | Health and social care workers | Motivational interviewing to promote oral health among homeless populations ('Smile4life programme') | Qualitative— telephone interviews, framework approach | Familiarity and good relationships between service providers and third sector organisations facilitated implementation, whereas lack of resources and interest hindered it | High quality |
| 2 | Beaton et al[26] (2018) UK | n=9 observation sessions, age not mentioned | Oral healthcare workers such as oral health educators and dental support workers | Motivational interviewing and tailored advice to promote oral health among the homeless population at different settings such as mobile dental units and homeless shelters ('Smile4Life programme') | Qualitative— participant observation, content analysis | Good working relationships between healthcare providers, patients and third sector organisations are important | Moderate quality |
| 3 | Beaton et al[27] (2021) UK | n=100, 16–85 years | Oral health practitioners, third sector organisation staff and local authority staff | Motivational interviewing and behavioural change techniques to promote oral health among the homeless ('Smile4Life programme') | Quantitative— questionnaire, K-R20, exploratory factor analysis, multivariate path analysis | Work practices such as positive attitudes and beliefs of the oral healthcare workers influence implementation | Moderate quality |
| 4 | Burnam et al[34] (1995) USA | n=276, mean age=37 years | Homeless individuals with co-occurring substance and mental health issues | Social model of residential and non-residential programmes providing integrated substance use and mental health services | Quantitative— structured interviews, regression analyses | Retention levels were higher in the residential programme compared with the non-residential one | Low quality |
| 5 | Coles et al[29] (2013) UK | n=14, age not mentioned | Healthcare workers from statutory and non-statutory organisations | A framework that offers tailored oral health advice and signposts to relevant dental services. ('Something To Smile About') | Qualitative— focus groups, content analysis | Oral health knowledge among the healthcare workers improved but complex needs such as housing, employment, etc, must be addressed prior to oral health for successful implementation | Moderate quality |
| 6 | Collins et al[35] (2019) USA | n=168, mean age=47 years | Homeless individuals with alcohol use disorder | Non-abstinence treatment programme that involves tracking of alcohol use, discussion of safe drinking practices and goal-oriented tasks ('Harm Reduction Treatment for Alcohol HaRT-A') | Quantitative— questionnaires, content analysis | It was positively viewed by the participants with high levels of retention and satisfaction | High quality |

Continued

**Table 2** Continued

| No. | First author (year) and country | Sample size and age of the participants | Participant group | Intervention description | Type of research, data collection and analysis | Findings of the paper relevant to the review | Quality appraisal/ ROB |
|---|---|---|---|---|---|---|---|
| 7 | Doughty et al[31] (2020) UK | Service users— n=353, age not mentioned Service providers—not stated | Homeless individuals and oral healthcare workers such as dentists, dental nurses, dental technicians, etc | Denture service provided by Crisis at Christmas Dental Service and Den-tech to the homeless and vulnerably housed | Qualitative | Communication, timing, resources and training were considered as areas that needed to be improved | Low quality |
| 8 | Forchuk et al[36] (2022) Canada | Service users— n=58, mean age=52.5 years Service providers— not stated | Homeless veterans with substance use problems and staff from housing services | Housing provided along with the peer support and harm reduction services to homeless veterans ('Housing First') | Qualitative— interviews and focus groups, thematic analysis | Stable housing with harm reduction services was well received. Collaboration between mental health and addiction services should be considered for future services | Low quality |
| 9 | Henderson[37] (2004) USA | Service users— n=15 Service providers— not mentioned | Homeless veterans with substance/ alcohol use and programme staff such as healthcare workers and administrative staff | Residential substance use treatment programme that focuses on relapse prevention along with education and housing stability for homeless men | Qualitative— surveys, direct observation and interviews, not stated | Majority of the participants provided positive feedback. Staffing issues such as training and competing workload were noted as drawbacks to the programme | Moderate quality |
| 10 | Neale and Stevenson[33] (2014) UK | Service users— n=30, 23–62 years Service providers— n=15, age not mentioned | Homeless individuals with substance use and mentors such as substance use workers, substance use managers and hostel staff | Computer-assisted therapies using 20 different psychosocial intervention strategies to identify and reduce substance use based in hostels and homeless shelters ('Breaking Free Online') | Qualitative— interviews, inductive coding and framework approach | 'Programme features', 'mentor support', 'participant characteristics' and 'delivery context' were noted as factors that lead to successful delivery | Moderate quality |
| 11 | Paisi et al[32] (2020) UK | Service users— n=11, 20–65 years Service providers— n=11, age not mentioned | Homeless individuals and the dental clinic staff members, support workers and volunteers | Community dental clinic that provides both regular and emergency treatments | Qualitative— semistructured interviews, reflective thematic analysis | Flexibility and the relationship between the patient and dental provider were highlighted as important features | High quality |

Continued

**Table 2** Continued

| No. | First author (year) and country | Sample size and age of the participants | Participant group | Intervention description | Type of research, data collection and analysis | Findings of the paper relevant to the review | Quality appraisal/ ROB |
|---|---|---|---|---|---|---|---|
| 12 | Pauly et al[38] (2020) Canada | n=14, 29–61 years | Homeless with illicit alcohol use | Non-residential community managed alcohol programme which provides harm reduction strategies and peer support ('Canadian Managed Alcohol Programme Study') | Qualitative—semistructured interviews, inductive coding and constant comparative analysis | Peer-led programme was successful as it facilitates capacity building, engagement, and empowerment | Moderate quality |
| 13 | Pratt et al[40] (2019) USA | n=40, 29–69 years | Homeless with smoking and alcohol use | Nicotine replacement therapy and motivational interviewing/cognitive behavioural therapy to reduce smoking and alcohol use among the homeless ('Power To Quit 2') | Qualitative—interviews, social constructivist approach to grounded theory | Social (peer groups) and environmental (housing, etc) factors impact cessation in homeless smokers | High quality |
| 14 | Pratt et al[41] (2022) USA | n=40, 29–70 years | Homeless with smoking and alcohol use | Nicotine replacement therapy and motivational interviewing/cognitive behavioural therapy to reduce smoking and alcohol use among the homeless ('Power To Quit 2') | Qualitative—semistructured interviews, social constructivist approach to grounded theory | Social pressure and shelter environment impact the intervention but the integrated treatment along with emotional support from the staff make it beneficial | High quality |
| 15 | Rash et al[39] (2017) USA | n=355, mean age=37 years | Homeless with substance use | Behavioural intervention contingency management with the use of incentives such as vouchers and prizes delivered at local community clinics | Quantitative—adaptation of the Service Utilisation Form, multivariate analysis of variance | Retention was higher in groups that accessed the intervention compared with the standard arm of care | Moderate quality |
| 16 | Rodriguez et al[28] (2019) UK | Service users— n=13, 18–22 years Service providers— n=5, age not mentioned | Young homeless people and NGO practitioners | Pedagogical workshops about oral health, mental health, substance misuse, diet, etc, to increase engagement and awareness | Qualitative—unstructured interviews and workshops, content analysis | Involvement of young people in co-designing an intervention facilitates engagement, trust building and increases health literacy | High quality |
| 17 | Stormon et al[30] (2018) Australia | n=76, 41–60 years Feedback— n=24 | Disadvantaged adults (clients of community organisations that use housing, employment and food services) | Facilitated access pathway between homeless organisations and public dental services. Improving oral health by assessing dental needs, offering dental advice and dental appointments | Quantitative—questionnaire, descriptive analysis, and framework approach | Positive feedback by participants facilitated by the environment, clinical staff and flexibility. Attendance rates varied across the site but was generally high. | Moderate quality |

ROB, risk of bias.

moderate quality,[26 29 33 37 38] due to reporting bias and five high quality.[25 28 32 40 41] The risk of bias was assessed for the five articles reporting quantitative findings; among the two RCTs, one had a high risk of bias because of attrition and reporting bias,[34] and the other article had a low risk of bias,[35] the remaining three cross-sectional studies were of moderate quality.[27 30 39] The findings reported in this review are mostly from high or moderate quality articles, with the inclusion of insights from low quality articles employed strategically for completeness of reporting evidence available and to supplement findings from the adequately reported articles.

### Synthesis of qualitative findings

Table 3 presents the themes, subthemes, codes and quotes from individuals experiencing SMD and frontline staff and stakeholders.

Synthesis of 12 papers with qualitative findings[25 26 28 29 31–33 36–38 40 41] identified three overarching themes in relation to the aims of this review. The three themes are (1) intervention settings, (2) intervention delivery and (3) ways to enhance engagement and participation.

### Theme 1: intervention settings

Eleven papers identified issues related to the settings of interventions which can play a role in the delivery of interventions targeting oral health, substance use and smoking.[25–28 32 33 36–38 40 41]

#### Physical settings

Physical settings involved the environment in which the intervention took place. The wider physical environment has been found to have an impact on the intervention experience,[41] with privacy being the key factor for improving physical settings.[32 33 41] Communal homeless shelters and busy teaching hospitals lack the space and privacy to deliver interventions involving discussions about difficult and sensitive topics.[32 33 41] Contrastingly, stable housing with the necessary privacy allowed people experiencing SMD to focus on their recovery journey, while also creating a space in which residents could spend time away from peers who were sometimes perceived as having a negative peer group influence.[36 40]

#### Psychological aspects of settings

Psychological aspects related to the less visible parts of the interventions were identified across 10 papers.[25 26 28 29 32 33 37 38 40 41] First, it was reported that relationships between people experiencing SMD and service providers played a vital part in the delivery of interventions. Through good communication,[26 28 32 41] trust building,[28 29 32 41] familiarity of working with a vulnerable population[25 32] and mentorship,[33 37] interventions were able to form a 'safe and respectable environment'.[28 32 37 41] Second, papers discussed the importance of peer support as a way of increasing the effectiveness of interventions.[32 37 38] There were also reports of the impact negative peer influence could have on the recovery process. For example, smoking and drinking were linked to socialising with others, which could increase the urge to smoke or drink.[40 41]

#### Accessibility

Accessibility of interventions was one of the factors found to be important related to implementation of interventions among people experiencing SMD.[25 26 32 41] First, accessible and spacious meeting points within the services were reported to help with their participation in the intervention, especially in the case of oral health interventions that were delivered either in a community setup (open space) or a mobile dental van .[26] Second, geographical proximity could act as a barrier as rural and remote areas lack the facilities and resources, which could influence the access of people experiencing SMD.[25 32] Lastly, it was reported that access could become an issue when service users move to more stable housing as weather conditions, distance, work and other appointments tend to make it challenging to attend the intervention sessions.[41]

### Theme 2: intervention delivery

Nine papers discussed aspects such as information availability, resources and perceived risks of working with a vulnerable population that could be important for roll out and delivery of interventions addressing oral health, smoking and substance use.[25 26 28 29 31–33 37 41]

#### Improved awareness

Awareness and information availability were discussed in papers focusing on improving oral health, smoking and alcohol use.[28 29 32 41] Sharing information between service providers and SMD groups was identified as an important issue across the papers as it created opportunities to promote involvement and behaviour change.[28 29 41] It was reported that easily understandable information encouraged people experiencing SMD to view healthier behaviours as important (eg, tooth brushing) and helped to signpost them to necessary services.[28 41] Clear and simple explanations of treatment options available was seen to help them in decision-making.[32] Service providers also felt that they learnt more about healthy behaviours and were able to pass their newly gained knowledge to their clients.[29]

#### Resources

Five papers discussed the importance of having necessary resources to enable interventions to run efficiently and effectively.[25 31–33 37] The majority of these highlighted the importance of distribution of workloads among staff because of difficulties in implementing interventions with competing duties and work within the organisations.[25 33 37] Funding and resources such as volunteers and materials were identified in oral health interventions as an important issue that impacts implementation and long-term sustainability.[31 32]

**Table 3** Themes and subthemes from qualitative synthesis of findings along with the relevant codes and quotes

| Themes | Subthemes | Codes | Quote(s)—people with SMD; or frontline staff |
|---|---|---|---|
| Intervention settings | Physical settings | Housing stability, privacy, confidentiality | "This is kind of a stressful situation. People are homeless, being at the bottom of their luck, and—boom—and everything. So this is stress. What do you do? You drink, and you smoke, and that's all that you can do, walking around here all day. Do you understand?" (person with SMD)[40] <br><br> "But at the same time the addictions piece, especially in terms of stability, I've noticed a lot of the guys that because they are stable in our home, they may make the choice more often to say 'I don't feel like drinking tonight,' so they don't. They don't have to get intoxicated to go to sleep in a shelter on a mat, they can choose not to drink and sometimes they do make that choice not to drink and just watch TV for the evening." (frontline staff)[36] <br><br> "If you went in and tried to do anything, people were behind you, over your shoulder, 'what are you doing there' And, you know, I didn't what to discuss with people what I was doing, because they'd take the mick." (person with SMD)[33] |
| | Psychological aspects of settings | Communication, trust building, familiarity, mentorship, community, peer pressure, guidance, support, safe space | "It's not just about dental treatment, I think for a lot of people there is the fear of the dentist because when they do go, it's because they need work done and they're in pain, therefore they associate pain with the dentist." (frontline staff)[29] <br><br> "She's not somebody that normally expresses much in a group, she's quite a private person, so I thought it took quite a lot for her to open up, to trust, but I also appreciate the fact that she felt she was in a really good space that she could share that experience with the others and I felt that was really valuable for the rest of the group to hear that. I think this activity [the workshops] encourages people to talk about their own experiences." (frontline staff)[28] <br><br> "Yes. A lot better off because… I'm not like, like when I'm here and I'm here with people that are drinking on programs like this and stuff like that I've noticed we're all on the same level. We don't care about the issues or problems, we just, you know, pitch together and do what we gotta do to get ourselves fixed and then from there if we can help other people, and people help other people…". (person with SMD)[38] <br><br> "If there is nobody there and you're just left to get on with it, it's quite easy to skip things…I will just put that answer down, you know. But then when you know somebody is there and they are there for that specific reason, then it's a lot easier to go through with things." (person with SMD)[33] |
| | Accessibility | Point of contact, space, geography | "We were put in the medical room along the corridor from the office, but there was no opportunity for practitioner 4 to approach any of the residents. We only saw service users if they specifically wanted to talk about their oral health or if they had walker past the room and wanted to see who we were." (frontline staff)[26] <br><br> "I'm from a very rural area, and we don't really have any homelessness centres." (person with SMD)[25] |
| Intervention delivery | Improved awareness | Understandable, ideas, learning from one and another | "Take things that people say and take it on board, and everything's a learning curve, you learn things all the time… And I'd recommend that to anybody else who is homeless, just listen to other people, take on board what they've got to say, and accept the help that's around you like the group activity [the workshops]." (person with SMD)[28] <br><br> "Especially when it was to do with what alcohol can do and what substances can do, I don't think they realized how that affects their oral health, their ears pricked up when you said that." (frontline staff)[29] |
| | Resources | Workloads, stress, competing needs, volunteers, equipment, funds | "I think he [client] felt that maybe I would have to sit with him again and, I don't know, maybe I should have sat him down and had a talk with him and I just haven't been able to." (frontline staff)[33] <br><br> "You feel like you're spinning so many plates, that you just can't possibly keep them all up in the air." (frontline staff)[25] <br><br> "We need to attract funding … it's very difficult to encourage NHS England to commission outside of their routine, the existing contract doesn't favour patients with high treatment needs so we would need them to step outside of their comfort zone and commission something slightly different to what they're used to." (frontline staff)[32] |
| | Perceived risks while working with a vulnerable population | Safety, unpredictable, inappropriate behaviour, challenges, relevant experience, confident, challenging behaviours | "Practitioner 1 is confident and appears quite fearless, putting up with language/ behaviour that would not be tolerated in a normal clinic." (researcher observation notes of frontline staff)[26] <br><br> "Initially we were thinking 'oh we need to make sure that we're not alone in the surgery at any point', and we had a panic alarm and things, we still have all that in place, but it's actually been fine." (frontline staff)[32] |

Continued

**Table 3** Continued

| Themes | Subthemes | Codes | Quote(s)—people with SMD; or frontline staff |
|---|---|---|---|
| Ways to enhance engagement and participation | Interest and motivation | Complexity, fears, initiative, specific and complex needs, mixed opinions | "(Mentor) came in and said 'I'm going home, have you done much?' And I said, 'I couldn't get back on, you know'. And she just took it [the laptop]. I don't know if she was fed up with me or whatever, but she never spoke about it again and I never mentioned it again." (person with SMD)[33]<br>"The oral health team do not seem bothered to recruit any patients, even if that means sitting waiting with nothing to do—the feeling seems to be that if a patient wants to be seen then they will come to the MDU." (researcher observation notes of frontline staff)[26]<br>"My goal is to quit within a month or two months. I talked to a couple of people. 'It ain't going to happen.' I said, 'well if you set your mind to certain things, you can do this." (person with SMD)[40]<br>"I think it's good. It made me feel like I had something to do or like I had a purpose. You know what I mean, not a purpose but it wasn't like the homeless."[41] |
| | Adapt to specific circumstances | Context, tailored to the needs of the individual, personalised care | "People getting through the door, they might not have a roof, might not have any money, might have major drug and alcohol issues, might be threatened with violence, the last thing they want to talk about is their teeth." (frontline staff)[29]<br>"We call these people chaotic and that's a bit judgmental, they are actually setting priorities, they've got so much going on in their lives that it [oral health] just falls of their list of priorities, they're saying 'it's my priority to find somewhere to sleep tonight' … The time that you catch people' was therefore identified as 'really important'." (frontline staff)[32] |
| | Constant support | Long-term care, advice, support | "About three or four in the morning and I feel like upset then…I can come down and use the program, which is quite good because that way I can put stuff that is all jumbled up in my head down in a way that makes sense and it kind of makes you see that things aren't quite so bad as they seem." (person with SMD)[33]<br>"At the stage of having goals, an action plan and were working through that… but for some homeless people who are nowhere ready, you can make an average of seven appointments before they will turn up once, it's just where your client is at." (frontline staff)[29] |

SMD, severe and multiple disadvantage.

### Perceived risks working with a vulnerable population

Papers reported on the perceived risks of delivering interventions to vulnerable populations as challenging at times by service providers.[25 26 32] There were concerns about safety of service providers while interacting with clients who were seen to be 'unpredictable'. The need for training and being better equipped to work in this environment and setting boundaries between service providers and clients was repeatedly mentioned by service providers.[25 26 32 37] The papers also highlighted the importance of training opportunities that provide service providers with the necessary skills to handle volatile and difficult situations.[25 37]

### Theme 3: ways to enhance participation and engagement

Ten papers identified factors such as interest and motivation levels, adaptability and long-term support that could help to improve outcomes and create sustainable interventions by enhancing engagement and participation.[25 26 28 29 31–33 37 40 41]

### Interest and motivation

Nine papers highlighted that the interest and motivation levels of both staff supporting SMD groups and people experiencing SMD play an important role in the implementation of interventions. Disinterest was sometimes observed among service providers, due to concerns about the complexity of delivering the intervention,[25 29 31] lack of engagement with third sector organisations,[26] poor uptake of the intervention by the target populations[25 29 31] and preconceived notions of improper behaviour by SMD groups.[41] Interestingly, interventions were met with similar feelings of indifference by people experiencing SMD if the intervention did not address their specific and complex needs such as housing and financial problems.[25 26 32] Two papers on oral health interventions found that younger adults and families with children were more eager to engage compared with single men.[28 29] Papers discussing the same smoking intervention illustrated that an awareness of health benefits and risks played a part in motivating people in engaging with the intervention.[40 41]

### Adapting to specific circumstances

Adaptability of interventions was noted as an essential feature among four papers.[29 32 33 40] Tailoring the interventions to address their specific needs at the time such as housing and employment was noted to increase participation and better outcomes.[29 40] Service users of a community dental service also suggested flexible and longer dental appointments would be helpful and in the long-term these adaptions would help reduce missed appointments.[32] Another paper reported that people experiencing SMD were keen to have more face to face

interactions rather than digital, which highlights the drive to more personalised care.[33]

### Long-term support

Four papers identified sustained and long-term support as a factor that could contribute towards better intervention outcomes.[29 32 33 37] Service providers expressed a need for interventions which allowed people experiencing SMD to continue with services/programmes despite missing appointments or not completing treatment within the required delivery timeframe especially because of the transitionary nature of SMD groups.[32 33] Similarly, for a substance reduction intervention, there is a preference for a long-term intervention, which allowed and supported them to gradually integrate into the new stage of their lives.[33 37] Two papers on oral health interventions suggested that drop-in services offered flexibility in seeking advice or seeing a practitioner and helped to reduce anxiety surrounding accessing treatment for dental health.[29 32]

### Synthesis of quantitative findings related to retention and implementation

Four papers reported quantitative findings on retention and programme attendance,[30 34 35 39] as indicators of uptake and sustained implementation of interventions.

Three papers on substance use interventions reported high levels of retention in their intervention groups.[34 35 39] Two studies among them delivered the interventions along with housing services but the findings were mixed and limited on whether retention was significantly associated with the housing services or not.[34 35 39]

There was no difference in the attendance levels in the studies related to substance use interventions.[34 35] The attendance level for an oral health promotion intervention delivered in community settings was high (85%); however, it varied across community centres and was dependent on timing of appointment and dental treatments offered. More non-attendance was seen for afternoon appointments and complex dental treatments (eg, surgical and prosthodontic treatments).[30]

Additionally, workplace beliefs and practices among service providers, such as knowledge, intention and goals, were reported to influence implementation behaviours.[27]

## DISCUSSION

This review synthesised different factors that could influence the implementation and sustainability of interventions related to improving oral health and related health behaviours of people experiencing SMD. Evidence suggested that psychological aspects of intervention settings such as building trust and communication form an integral part in the creating a safe environment and that these are just as essential as the structural components of settings such as physical environment. Review findings further suggest that adequate staff capacity, funding and equipment would ease the delivery of interventions by reducing the immense pressure faced by service providers supporting the interventions. It was also suggested that implementation is dependent on the interest and motivation of not only people experiencing SMD but also on that of service providers in delivering difficult and complex interventions.

Most of the included studies were related to oral health and substance use (drug and alcohol). There was a lack of evidence on diet and smoking interventions among this population. Previous evidence has shown that tobacco use and poor diet, often due to limited choice available while experiencing homelessness and related disadvantages, result in a range of adverse short-term (nutritional deficiencies) and long-term health outcomes (cancer, diabetes, heart disease).[42–44] Food insecurity is often linked to elevated tobacco use, mental health issues and an increased risk of substance misuse.[45–47]

While most of the papers mainly focused on the perspectives of people experiencing SMD, the limited data from service providers brought light to some of the challenges faced during implementation. This supports the notion that intervention implementation needs the coordination and collective effort of everyone involved. All the interventions included were designed focusing on service provision,[25–28 30–35 37–41 48] except for one study that focused on a training intervention for service providers.[29] Limited evidence was available on the long-term sustainability of interventions, which highlights another evidence gap that needs to be addressed.

Our review findings suggest that the retention in interventions may depend on the type of treatment offered, which at times can be influenced by the availability of housing provision. Timing and type of treatment may also influence attendance rates; for instance, morning appointments might be more beneficial, especially for individuals struggling with alcohol addiction, as they may be less intoxicated compared with later in the day. Our review findings also complement our systematic review about the effectiveness of interventions that improve oral health and related health behaviours in SMD groups—the effectiveness review found that interventions that integrated health with the individual's wider needs (eg, housing, employment, mental health) were more effective than usual care.[49] The findings we have are very limited regarding retention and attendance, more effort needs to be taken to understand how to improve reach and retention among SMD groups so that they can access and use the interventions efficiently.

A systematic review on access to dental care among individuals experiencing homelessness in the UK identified similar findings around awareness, accessibility and organisational issues (lack of financial resources and collaboration between sectors) having an influence on implementation.[50] This was also similarly identified in another review on smoking cessation among homeless populations in high-income countries.[51] The importance of continued engagement in services was highlighted in a review on substance use support for young people (ages 12–24) experiencing homelessness, which was also reflected in our findings.[52] Existing literature

on interventions targeting health conditions such as HIV and hepatitis C in this population have shown that improved health outcomes are linked to increased awareness, establishment of positive relationships with service providers and integrated treatment involving other health behaviours.[53–55]

Some findings from our review on aspects related to intervention settings and intervention delivery aligned with CFIR constructs of inner and outer settings domains.[14] Subthemes in our findings on ways to enhance engagement aligns with both individuals and implementation process domains.[14] The use of CFIR framework helps us understand the impact of intervention settings, delivery methods and engagement on the implementation process. It also provides a comprehensive approach for guiding the development of interventions targeting SMD groups and improving their efficacy in practical settings.

## Strengths and limitations

This systematic review is novel in that it assesses the implementation and sustainability of interventions on oral health together with co-occurring and related health behaviours in people experiencing SMD. It addresses an evidence gap on interventions targeting these health challenges and identifies ways to overcome implementation issues faced by these specific interventions. Another strength of this review lies in its comprehensive search strategy and use of a published tool (ie, CFIR) to make sense of the results. It also highlights gaps in the evidence base on interventions related to diet, as well as studies that include repeat offenders. However, the confidence in the evidence from this review is limited as most of the papers were of moderate quality. Studies lacked detailed data collection methods and standardised evaluations which influenced their quality. Another limitation of this work is that intersectionality was not considered explicitly during the analysis of the data. Furthermore, the findings may not be generalisable to all contexts since the included papers were from high-income countries.

## Implications

These findings offer valuable insights for enhancing existing interventions by paying attention to settings, delivery and engagement opportunities. Evidence from this review points to the need for additional research on interventions targeting smoking and diet. These areas hold significant value due to their direct links with general and oral health. It is also important for interventions to address not only individual behaviours but also overlapping behaviours of substance use, smoking and poor diet. This could help reduce the strain on resources and improve engagement. Furthermore, higher quality research that focuses more on sustainability and intersectionality is warranted to further investigate and refine interventions focused on SMD groups.

## Author affiliations
¹Population Health Sciences Institute, Newcastle University, Newcastle upon Tyne, UK
²Evidence Synthesis Group and Innovation Observatory, Population Health Sciences Institute, Newcastle University, Newcastle upon Tyne, UK
³NHS England and NHS Improvement, Newcastle upon Tyne, UK
⁴Department of Epidemiology and Public Health, University College London, London, UK
⁵NIHR Policy Research Unit Behavioural Science, Population Health Sciences Institute, Newcastle University, Newcastle upon Tyne, UK
⁶Department of Public Health, Social and Preventive Medicine, Centre for Preventive Medicine and Digital Health (CPD), Heidelberg University Medical Faculty Mannheim, Mannheim, Germany
⁷Faculty of Medicine and Dentistry, Peninsula Dental School, Plymouth University, Plymouth, UK

**Contributors** Conceptualisation: SER, EK, EAA, CB, RGW, ECJ, LJM, FRB, DC, DL, MP, FFS. Methodology: SER, DAJ, LJMG, EAA, ECJ, CR, EK, DC, FRB. Draft preparation: SER, DAJ, EAA. Writing—review and editing: All authors. Funding acquisition: SER, EK, FRB, CB, RGW, DC, DL, MP, FFS. All authors contributed to the interpretation of data and the final version of the manuscript, and all are guarantors.

**Funding** This project is funded by the National Institute for Health and Care Research (NIHR) Policy Research Programme (NIHR200415). EAA was supported by the National Institute for Health and Care Research (NIHR) School for Public Health Research (SPHR) Pre-doctoral Fellowship, Grant Reference Number PD-SPH-2015. Now, EAA (Doctoral Research Fellow, NU-010978) is funded by the NIHR for this research project. EK is supported by an NIHR Senior Investigator award and directs the NIHR funded Applied Research Collaboration North East and North Cumbria. SER, EK, EAA, LJM, DAJ are members of Fuse, The Centre for Translational Research in Public Health (www.fuse.ac.uk) and NIHR Applied Research Collaboration North East North Cumbria. Fuse is a UK Clinical Research Collaboration (UKCRC) Public Health Research Centre of Excellence. Funding for Fuse from the British Heart Foundation, Cancer Research UK, National Institute of Health Research, Economic and Social Research Council, Medical Research Council, Health and Social Care Research and Development Office, Northern Ireland, National Institute for Social Care and Health Research (Welsh Assembly Government) and the Wellcome Trust, under the auspices of the UKCRC, is gratefully acknowledged. FFS and EK are Senior Investigators in the NIHR Policy Research Unit in Behavioural Science.

**Competing interests** None declared.

**Patient and public involvement** Patients and/or the public were not involved in the design, or conduct, or reporting or dissemination plans of this research.

**Patient consent for publication** Not applicable.

**Ethics approval** Not applicable.

**Provenance and peer review** Not commissioned; externally peer reviewed.

**Data availability statement** No data are available.

**ORCID iDs**
Deepti A John http://orcid.org/0000-0003-3969-7821
Emma A Adams http://orcid.org/0000-0001-7536-0658

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
