## [Reviewer comments · BMJ Open]

ARTICLE DETAILS

TITLE (PROVISIONAL)	Factors influencing implementation and sustainability of interventions to improve oral health and related health behaviours in adults experiencing severe and multiple disadvantage: A mixed-methods systematic review
AUTHORS	John , Deepthi; Adams, Emma; McGowan, Laura; Joyes, Emma; Richmond, Catherine; Beyer, Fiona; Landes, David; Watt, Richard; Sniehotka, Falko; Paisi, Martha; Bambra, Clare; Craig, Dawn; Kaner, Eileen; Ramsay, Sheena

VERSION 1 – REVIEW

REVIEWER	Henson, Connie Macquarie University Faculty of Human Sciences, Health Sciences
REVIEW RETURNED	25-Oct-2023

GENERAL COMMENTS	Thank you for asking me to review this excellent manuscript. It is well-written and has been conducted in a responsible manner. The incorporation of the CFIR and quality tools is reassuring. My one fundamental question is why the authors choose to include the low-quality studies and consider/weigh that evidence equal to the high-quality studies. Could you make stronger statements if you report on all findings but highlight the findings specifically from the high-quality studies? Minor suggestions Line 26 change 'an' to 'a' Line 48 'being important towards implementation,' sounds a bit awkward and vague to me – could it be 'negatively influenced implementation'? Line 33 – do you mean a limitation was that intersectionality was not considered explicitly during the analysis of the data?
---

REVIEWER	Benton, Erin Winston-Salem State University, Exercise Physiology
REVIEW RETURNED	02-Nov-2023

GENERAL COMMENTS	For the articles you included in the review (17), the majority of them have been published in the past 10 years. However, article 4 (Burnam et al, 1995) is significantly older than the others and labeled low quality with a high risk of bias. I would consider removing this article from the review. It does not make sense that it is included and does not add anything significant.
---

VERSION 1 – AUTHOR RESPONSE

Reviewer 1 Comments:

	Comments	Response to the comment	Changes made in revised manuscript
1.	Thank you for asking me to review this excellent manuscript. It is well-written and has been conducted in a responsible manner. The incorporation of the CFIR and quality tools is reassuring.	We thank the Reviewer for this comment.	N/A
2.	My one fundamental question is why the authors choose to include the low-quality studies and consider/weigh that evidence equal to the high-quality studies. Could you make stronger statements if you report on all findings but highlight the findings specifically from the high-quality studies?	We thank the Reviewer for this comment. We appreciate the Reviewer’s comment about the inclusion of ‘low-quality’ studies and agree that further justification is needed about inclusion of studies. In the Methods section, we have now provided further justification to explain that poor reporting is not always reflective of poor methodology. We included all studies that met the inclusion criteria to allow capturing all the available evidence on our review objectives; this has allowed us to report the global evidence available related to implementation and delivery of interventions to improve oral health and related behaviours in people experiencing SMD. As the	As suggested, we have now added a clearer justification and have highlighted the point about findings based on high-quality studies – Methods section, (Data extraction and quality appraisal - page 5) and Results section (page 14).

	Comments	Response to the comment	Changes made in revised manuscript
		Reviewer suggests, we have been mindful of the impact of low-quality studies while reporting findings. The main findings take account of the quality of evidence and are based on high-quality studies.	
3.	Line 26 change 'an' to 'a'	We have edited this as suggested.	As suggested, we have corrected the typo on page 26, paragraph 2.
4.	Line 48 'being important towards implementation,' sounds a bit awkward and vague to me – could it be 'negatively influenced implementation'?	We thank the Reviewer for this suggestion. We accept the need to re-word the sentence to make it clearer to the reader.	As suggested, we have edited the sentence in the Discussion section (page 27, paragraph 1).
5.	Line 33 – do you mean a limitation was that intersectionality was not considered explicitly during the analysis of the data?	We thank the Reviewer for bringing this typo/error to our attention. We have corrected the sentence to say that intersectionality was not considered during the data analysis.	The sentence has been edited, as suggested in the Discussion section (Strengths and Limitation - page 27).

Reviewer 2 Comments:

	Comments	Response to the comment	Changes made to the manuscript
1.	For the articles you included in the review (17), the majority of them have been published in the past 10 years. However, article 4 (Burnam et al, 1995) is significantly older than the others and labeled low quality with a high risk of bias. I would consider removing this article from the review. It does not make sense that it is included and does not add anything significant.	We thank the Reviewer for this suggestion. We acknowledge that the paper by Burnam et al, 1995 is an older paper and has been classed as 'low-quality' in the quality appraisal. We included all studies in the review to gather the full global evidence on our review questions, to allow reporting completeness of evidence. We accept that there are limitations in this paper. In agreement with the Reviewer, we have now pointed out in the Results section that this was a low-quality study and	We have clarified the use of low quality studies within this review – Methods section, (Data extraction and quality appraisal - page 5) and Results section (page 14).

		was reported for completeness of evidence found and to supplement the results. We agree that further justification and clarification would be helpful in the Methods section, which we have now added.	
--	--	--	--